# Tidal Flushing Rather Than Non-Point Source Nitrogen Pollution Drives Nutrient Dynamics in A Putatively Eutrophic Estuary

**Johannes R. Krause** [1,*] , **Michelle E. Gannon** [2] , **Autumn J. Oczkowski** [3] , **Morgan J. Schwartz** [3] , **Lena K. Champlin** [2] , **David Steinmann** [4] , **Martha Maxwell-Doyle** [5] , **Emily Pirl** [5] , **Victoria Allen** [2] and **Elizabeth Burke Watson** [2]

1    Coastlines and Oceans Division, Florida International University, Miami, FL 33199, USA
2    Department of Biodiversity, Earth & Environmental Sciences and the Academy of Natural Sciences, Drexel University, Philadelphia, PA 19103, USA
3    US Environmental Protection Agency, Atlantic Coastal Environmental Science Division, Center for Environmental Measurement and Modeling, Office of Research and Development, Narragansett, RI 02882, USA
4    Hammonton MLRA Soil Survey Office, National Resources Conservation Service, Hammonton, NJ 08037, USA
5    Barnegat Bay Partnership, Toms River, NJ 08754, USA
*    Correspondence: jkrause@fiu.edu

**Abstract:** The effects of nonpoint source nutrients on estuaries can be difficult to pinpoint, with researchers often using indicator species, monitoring, and models to detect influence and change. Here, we made stable isotope measurements of nitrogen and carbon in sediment, water column particulates, primary producers, and consumers at 35 stations in the reportedly eutrophic Barnegat Bay (New Jersey) to assess N sources and processing pathways. Combined with water quality and hydrological data, our C and N isoscapes revealed four distinct geographic zones with diverging isotopic baselines, indicating variable nutrient sources and processing pathways. Overall, the carbon stable isotopes ($\delta^{13}$C) reflected the terrestrial-marine gradient with the most depleted values in the urban and poorly flushed north of the estuary to the most enriched values in the salt marsh-dominated south. In contrast, the nitrogen stable isotope values ($\delta^{15}$N) were most enriched near the oceanic inlets and were consistent with offshore $\delta^{15}$N values in particulate organic matter. Several biogeochemical processes likely alter $\delta^{15}$N, but the relatively lower $\delta^{15}$N values associated with the most urbanized area indicate that anthropogenic runoff is not a dominant N source to this area. Our findings stand in contrast to previous studies of similar estuaries, as $\delta^{15}$N signatures of biota in this system are inversely correlated to population density and nutrient concentrations. Further, our analyses of archival plant (*Spartina* sp., *Phragmites australis*) and shell (*Geukensia demissa*, *Ilyanassa obsoleta*) samples collected between 1880 and 2020 indicated that $\delta^{15}$N values have decreased over time, particularly in the consumers. Overall, we find that water quality issues appear to be most acute in the poorly flushed parts of Barnegat Bay and emphasize the important role that oceanic exchange plays in water quality and associated estuarine food webs in the lagoon.

**Keywords:** stable isotopes; eutrophication; isoscape; nitrogen fixation; flushing; non-point source; estuary

## 1. Introduction

The effect of anthropogenic nutrient enrichment on coastal ecosystems has been an important theme in coastal ecological research for more than 50 years, where excess nutrients, and in particular nitrogen (N), from sewage, agricultural drainage, and urban runoff have had a myriad of negative effects on our coastal ecosystems [1,2]. Nitrogen is the nutrient that most commonly limits production in estuaries, such that additional N increases productivity which can, in turn, support nuisance and harmful algal blooms that

can cause a number of problems; from shading ecologically valuable submerged aquatic vegetation to producing toxins that impact human health [3,4], to the enhancement of erosion in N-enriched tidal marsh creeks, reducing the resilience of coastal habitats to sea level rise [5]. Nitrogen-supported eutrophication is also known to reduce water column oxygen concentrations as the organic matter breaks down [2]. Numerous efforts to reduce N loads have been successful and cases of successful recovery of coastal ecosystem function after reducing N-loads are becoming more common [6–8]. Along the Atlantic coast of the US, many point source reductions have been undertaken, at least in larger systems, with sewage treatment plants associated with cities discharging offshore [9,10] or upgrading to advanced sewage treatment with N removal [11]. While these improvements are costly, the timing of the changes is clear. However, in many estuaries, most of the nutrients are entering the estuary through nonpoint pathways like groundwater underflow and surface water runoff, including rivers and streams, which can collect diffuse flows and discharge them efficiently to coastal water bodies. These flows are much more difficult to assess and remediate, but are equally important to address [12,13].

Quantifying the effects of nonpoint source nutrients on coastal habitats and tracking the effects of changes on receiving water bodies is a research challenge [12,14]. Characterizing sources consumes time and resources. For example, overland flows are episodic, generally associated with storm events, and groundwater contributions can be diffuse and difficult to characterize amongst coastal aquifers [15]. Atmospheric deposition can be important and has clearly declined over time [16], but accurate quantification requires multiple nearby monitoring stations collecting long-term data. This is not found for most estuaries [17]. It is even harder to track changes in nonpoint source pollution given the challenges associated with timing and delivery. For example, it may take years to decades for groundwater to move from source to sink [14]. However, concerted efforts to reduce nonpoint source nutrient loads can lead to positive downstream ecosystem responses [14], and researchers have used indicator species [18,19], water quality monitoring [14], and models [16] to identify and assess responses to nonpoint source pollution reductions.

Stable isotopes are often used to assess coastal N sources and sinks [20,21]. They can provide important insights into nutrient concentrations, sources, and processing over time. Stable isotope values are typically reported as:

$$X‰ = \frac{X_{sample} - X_{standard}}{X_{standard}} \times 1000‰ \tag{1}$$

where $X$ is the ratio of the heavy isotope to the light isotope (i.e., $^{15}N/^{14}N$, $^{13}C/^{12}C$). Particularly stable isotopes of nitrogen (N, $\delta^{15}N$) and carbon (C, $\delta^{13}C$) are frequently used, in conjunction with other ecological information, to determine relative contributions of different N and C sources as well as indicate levels of N cycling and ecosystem productivity [22,23]. Depending on the trophic level and water quality gradients, estuarine stable isotope values can reflect different sources or processes. The $\delta^{15}N$ values increase with trophic level, and can be used to infer trophic structure, but residence time also increases $\delta^{15}N$ [24]. While $\delta^{13}C$ values fractionate only slightly with trophic level, they vary as a function of C source and assimilation pathway (C3, C4, CAM; [20]). Phytoplankton are generally characterized as having $\delta^{13}C$ values of about $-22‰$, but higher water column productivity, like during a phytoplankton bloom, will result in less negative $\delta^{13}C$ values [20,22]. Unlike most other biogeochemical measurements, isotopes are a ratio (heavy: light isotopes) of ratios (sample: standard) [20]. While concentrations of individual isotopes are quantifiable, their ratios, expressed in per mil (‰) units, are most often used as indicators of sources and sinks. Given the dynamic nature of the isotope values, and the various processes influencing their composition, they are best considered in conjunction with other ecological information.

Despite their complexities, there is a large body of literature demonstrating that stable isotopes can track anthropogenic nutrients. For example, high $\delta^{15}N$ values from sewage treatment plants, before, during, and after upgrades, were measured in the Narragansett

Bay food web, with values changing as sewage source values changed [22]. More broadly, others have used cross-systems comparisons to illustrate how higher $\delta^{15}$N values were related to upstream landuse and other nutrient sources across estuaries [23,25], including septic inputs [26] and food sources within an estuary (e.g., [27]). While isotopes are used as tracers of specific sources, they can also be thought of more broadly as tracking water masses. In the example from Narragansett Bay, $\delta^{15}$N in water particulates and macroalgae reflect a gradient as water masses from the Providence-Seekonk River Estuary mix with shelf water over the length of the Bay [11]. Isotope measurements in macroalgae have proven to be sensitive indicators. Given the challenges of assessing nonpoint source nutrient impacts on coastal systems and variations in food sources and in situ processing, stable isotopes have been proven to be an effective tool to look for spatial and temporal patterns in water quality impairment.

Because coastal systems are changing over long periods of time, with shifting nutrient inputs, warming waters, and other effects of climate change, researchers have developed and refined methods to measure the $\delta^{15}$N values of shell-bound organic material, allowing them to glean information about historical N sources and sinks from archival samples [28–30]. These values serve as a record of $\delta^{15}$N of suspended particulate matter (SPM) [31,32] and provide information about N sources to the mollusk which reflect local nutrient processing [33]. Shells from museum collections, archives, and excavations from dated middens allow us to develop timelines of N sources and cycling that extend back thousands of years [22,31]. Similarly, archival plant clippings and preserved fish have been used to track historical changes in atmospheric $\delta^{13}$C associated with fossil fuel emissions [34], as well as human impacts on coastal fisheries [35].

The Barnegat Bay-Little Egg Harbor-Great Bay (BB-LEH-GB) lagoon system comprises a compelling case study for the use of stable isotopes as a tool to assess nonpoint source nutrient pollution as there is a well-documented gradient from high nutrient loads and low flushing in the north of the estuary, to lower loads and shorter residence times further south, by the lagoon's inlet [36–39]. While the northern part of BB-LEH-GB is heavily developed, there are no point sources of nutrients as wastewater has been collected and discharged 1.6 km offshore since the 1980s [38]. However, local managers have observed periods of lower dissolved oxygen and high turbidity, in violation of current water quality standards, as well as a decline in seagrass and clams and an increase in episodic algal blooms [39]. In their assessment of the water column nutrient data collected as part of the New Jersey Department of Environmental Protection's monitoring program, Pang et al. [39] concluded that there was a need for site-specific water quality targets for BB-LEH-GB, to better reflect the spatial variation in the system and allow for more targeted restoration efforts.

Building on this recommendation, we used stable isotopes to characterize the macronutrient landscape in the BB-LEH-GB system to both identify where problems may be and to provide context for potential future changes, such as with proposed tide gate installations [40]. Key questions addressed by this study included the following: first, we aimed to determine how macronutrient (C, N) isotope ratios varied with nutrient loads and hydrodynamic characteristics at BB-LEH-GB, including the reportedly eutrophic northerly Barnegat Bay [38,39]. Secondly, we assessed whether the N isotope composition of biota has increased over time with watershed population increases, as has been found in nearby estuaries such as Long Island Sound and Narragansett Bay [23,41]. Lastly, as our data pointed at the importance of oceanic exchange as a driver of both spatial and temporal isoscape (mapping of isotope distributions) patterns, we provided context on spatial and temporal shifts in tidal flushing. Overall, we found spatial patterns in C, N isotope composition that reflect contrasts in macronutrient sources and processing. In concert with paired analysis of plant and shell material from museum and modern collections, our isoscapes suggest that challenges to water quality at Barnegat Bay are related to flushing, rather than anthropogenic nonpoint source nutrient pollution.

## 2. Materials and Methods

### 2.1. Study System

The BB-LEH-GB is a shallow back-barrier tidal lagoon extending nearly 70 km from the northern Metedeconk River (40.07° N, −74.05° W) to southern Great Bay (39.49° N, −74.31° W) in New Jersey. The lagoon has a surface area of 280 km$^2$ and an average depth of 1.5 m with extensive areas less than 0.5 m [42]. The BB-LEH-GB currently has three inlets: Little Egg Inlet to the south, Barnegat Inlet in central BB-LEH-GB, and Manasquan Inlet in the north. The tidal range of the BB-LEH-GB estuary is restricted by the shallow depth of the lagoon, and ranges from 0.5–1 m at Little Egg Inlet, 1.4–1.5m at Barnegat Inlet, to minimal at Manasquan Inlet [42], and tides attenuate landward to 0.2 m [36].

As many as six inlets allowed for tidal exchange during the 19th century. Cranberry Inlet, located between Barnegat and Manasquan, was open from 1758 to 1812, and Beach Haven Inlet was open north of Barnegat Inlet in the 1840s to 1860s and again in the 1920s (Table S1). In addition, two inlets (called New Inlet and Old Inlet) were open at various time points at Little Egg Harbor (Table S1). The tidal prism of the estuary has varied during the 20th century and increased about 50% between the 1930s and 1940s in response to coastal engineering projects which widened and stabilized Barnegat Inlet, then gradually decreased to one third of 1940s values by the 1980s due to sedimentation [42]. Between 1987 and 1991, the US Army Corp of Engineers dredged and realigned the inlet, which increased tidal exchange to 1930s levels [42]. Mean residence time of BB-LEH-GB has been estimated at 9–13 days by Defne and Ganju [36], but substantial spatial variability was identified. Daily flushing near the current oceanic inlets is contrasted by residence times up to 30 days in southern Little Egg Harbor and in the north of Barnegat Bay near Toms River [36].

The BB-LEH-GB watershed spans approximately 3500 km$^2$ [43] and is largely coincident with Ocean County. Currently, land use in Ocean County is about 35% developed, 63% forested, and 2% agricultural [44], excluding areas of open water and tidal marsh (Figure 1). Although water column N levels are observed to be quite low [37], BB-LEH-GB has been classified as a highly eutrophic system based on criteria of the National Oceanic and Atmospheric Administration's National Estuarine Eutrophication Assessment related to excess chlorophyll levels [45]. Nitrogen enrichment has been associated with a number of negative effects, including algal growth, harmful algal blooms, high turbidity, bacterial pathogens, and impacts to fisheries, seagrass, and shellfish beds [19]. Wastewater has been discharged 1.6 km offshore since the 1980s [38], and in 2011, the nation's strongest fertilizer bill was passed in New Jersey, which set limits on fertilizer content, requires application by a certified landscape professional, and restricts N application (Figure 1; [46]).

### 2.2. Sample Collection

Flora, fauna, soil and water samples were collected from the BB-LEH-GB system during the summers of 2019 and 2020, at 35 assessment points located in tidal wetlands (Figure 2; Table S2).

Soils (0–5 cm), *Spartina alterniflora*, *Phragmites australis*, and *Spartina patens* were collected at five widely spaced locations (10 m + apart) and mixed to represent a site average. Water was obtained by a grab sample from each of 25 sites and filtered through a precombusted 0.7 μm glass fiber filter to isolate suspended particulate matter (SPM). Crabs were collected after 24-hr deployments of pitfall and quarter size blue crab traps outfitted with turtle excluders (baited with Atlantic menhaden). Fish were collected in minnow traps placed approximately 10 to 20 m apart. Mud snails (*Ilyanassa obsoleta*), *Zostera marina* and *Ruppia maritima* were collected opportunistically and were not present at all sites (Figure 2). *R. maritima* was found in ponds on the marsh surface, while *Z. marina* was collected from sub-tidal areas adjacent to the marsh.

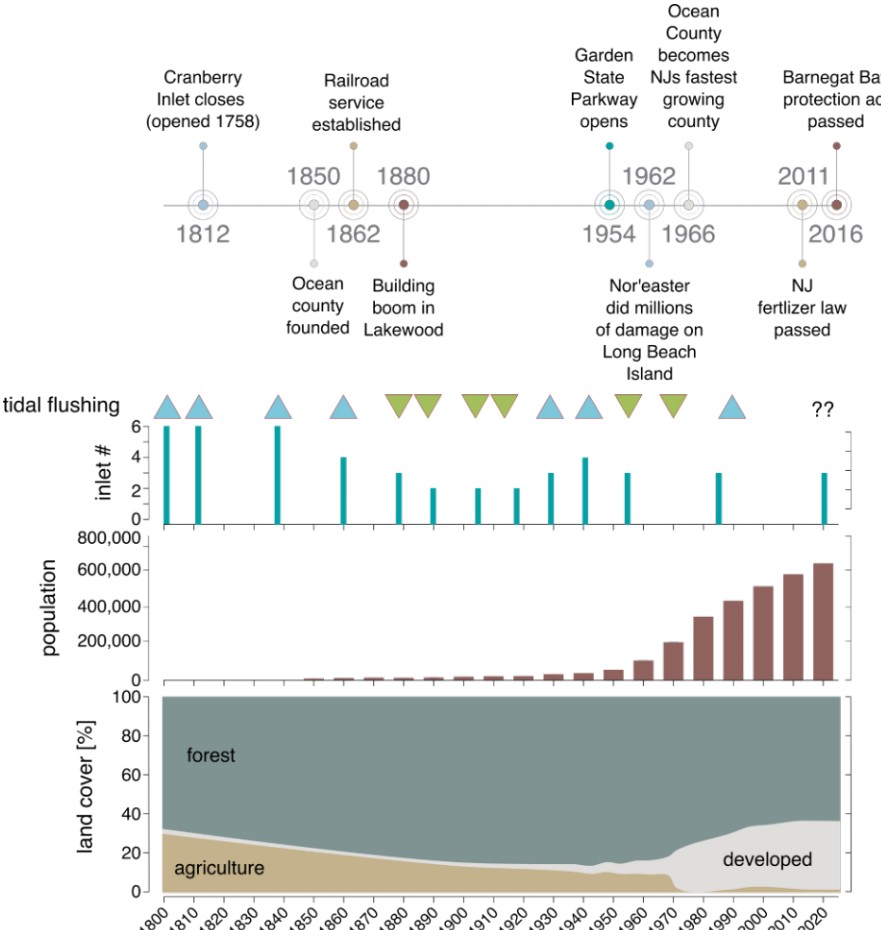

**Figure 1.** Timeline of Ocean County events, population, and landcover 1800–2020. Estimated tidal flushing of Barnegat Bay is depicted as greater (up arrow, in blue) or lesser (down arrow, green). Tidal flushing was assessed qualitatively as the number of tidal inlets present, as well as based on 20th century tidal prism measures. Ocean County is largely coincident with the Barnegat Bay-Little Egg Harbor-Great Bay (BB-LEH-GB) system and its watershed. Landcover excluded open water and tidal wetlands. Data sources include US census, agricultural census, USGS, and NJ DEP, and historic maps for tidal prism measures (Table S1). Nodes on the landcover map were auto-smoothed.

Soil cores were collected using a vibracore tube and analyzed for soil characterization by the United States Department of Agriculture (USDA)—Natural Resources Conservation Service soil scientists. The vibracore samples were obtained using a 3-inch (5.1 cm) diameter aluminum vibracore tube that was drilled into the soil using a power-driven vibrating head attached to the tube while applying constant downward pressure. Soil cores were drilled to a maximum of 2 m or until refusal. The vibracore tube was then filled with water and capped tightly to create a vacuum, and the tube was extracted utilizing a chain hoist attached to a ladder. The vibracore tubes were split, described, sampled, and classified utilizing National Cooperative Soil Survey Standards [47].

Historical samples archived in museum collections were analyzed with permission for destructive sampling (Table S4). Mollusks were obtained from the Academy of Natural Sciences of Drexel University (ANS), the Delaware Museum of Natural History, and American Museum of Natural History and range in collection years between 1880 and 1975. Plant samples were obtained from the New York Botanical Gardens and the Carnegie Museum of Natural History and range in collection dates between 1890 and 1946. Modern samples were collected to match those sampled from museum collections, including *I. obsoleta*, *Mercenaria mercenaria*, *Mya arenaria*, *Crassostrea virginica*, and *Spisula solidissima*.

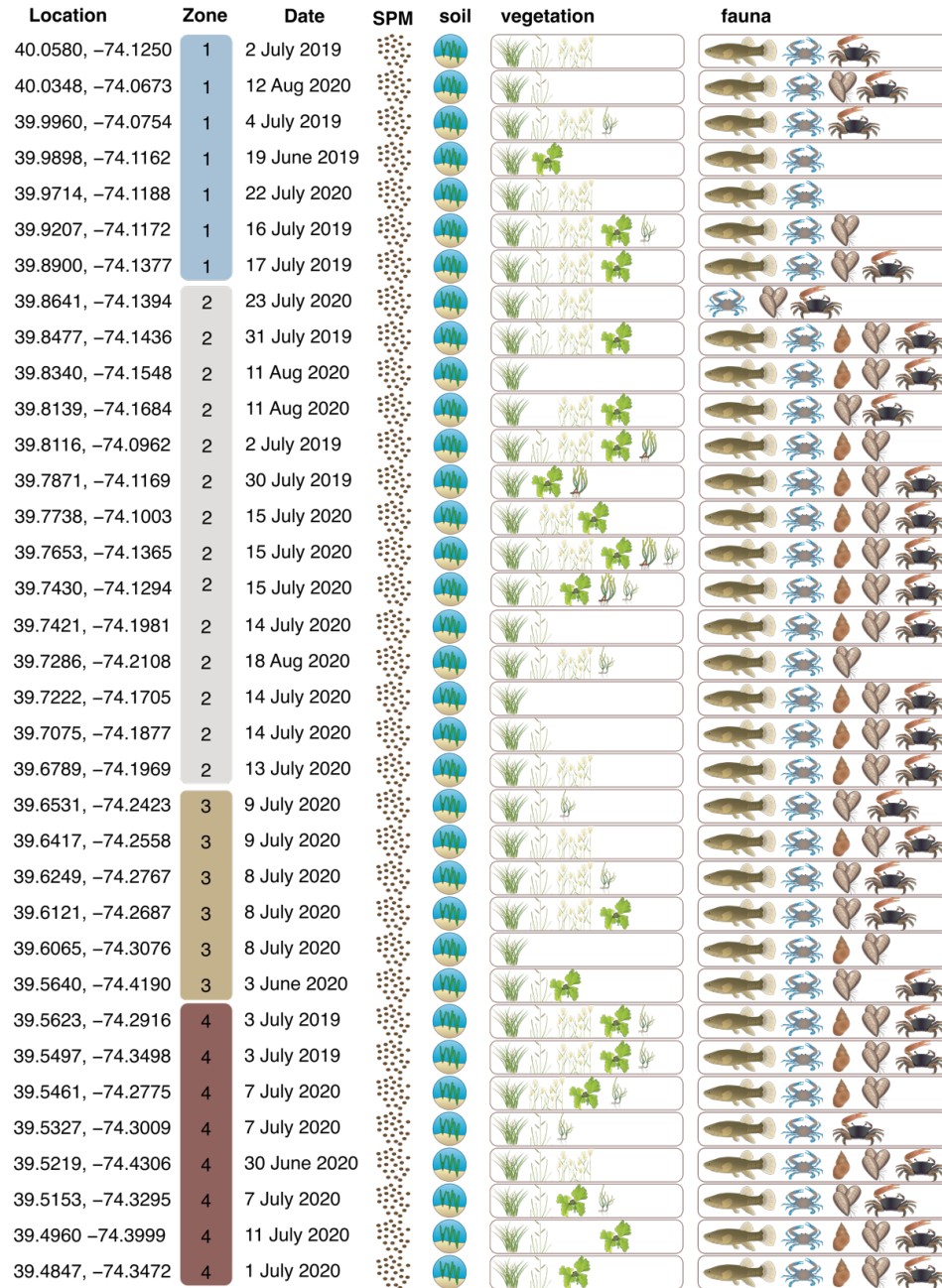

**Figure 2.** Samples collected and analyzed for stable isotopes of carbon and nitrogen. Plant species from left to right include *Spartina alterniflora*, *Spartina patens*, *Phragmites australis*, *Ulva* spp., *Zostera marina*, and *Ruppia maritima*. Consumer species from left to right include *Fundulus* spp., *Callinectes sapidus* (blue crab), *Ilyanassa obsoleta* (mud snail), *Geukensia demissa* (ribbed mussel), and *Uca* spp. (fiddler crab). Images from The University of Maryland Center for Environmental Science Integration and Application Network (IAN; Table S3) (https://ian.umces.edu/media-library/ (accessed on 15 July 2022)). SPM stands for suspended particulate matter.

*2.3. Laboratory Analysis*

2.3.1. Stable Isotopes

Samples collected in 2019 (n = 10) were composited while those collected in 2020 were analyzed independently (n = 25) (Figure 2; Table S2). All animal and plant samples were rinsed with deionized water prior to processing. *Ilyanassa obsoleta* tissues were separated from the shell and tissues of the left cheliped of *C. sapidus* were isolated. *Fundulus* spp. were

fileted to remove bones and scales. To remove inorganic carbon, aliquots of *Fundulus* spp., *C. sapidus*, *Uca* spp. and soil samples were fumigated prior to analysis [48].

Samples were analyzed using an Elementar Pyrocube interfaced with an Elementar Isoprime100 Isotope Ratio Mass Spectrometer (at ANS) or with an Elementar Vision and Elementar Vario Isotope Select (at the EPA). Isotope values were calculated based on reference standards and in-house working standards which have a precision at or better than $\pm 0.40‰$ (N) and $\pm 0.14‰$ (C) based on long term replication.

Shells of museum and modern specimens were powdered with a handheld Dremel tool with a diamond bit. Values of $\delta^{15}N$ from shell-bound carbonate were measured at ANS with the instrumentation detailed above, however for this analysis, the Pyrocube moisture trap was retrofitted to include a section of NaOH to remove evolved $CO_2$ followed by sicapent$^®$ for the removal of evolved $H_2O$, allowing $N_2$ to enter the isotope ratio mass spectrometer [32].

### 2.3.2. Dissolved Inorganic Nutrients

Dissolved ammonium, nitrate-nitrite, and phosphate were measured from aliquots of water collected at each of the 2020 collection sites using an Astoria Pacific 2 continuous flow analyzer via US EPA methods 353.3, 350.1, and 365.1. Samples were calibrated against a six-point standard curve, check standards were run every 15 samples, and Milli-Q blanks were run every 10 samples.

### 2.3.3. Salt Marsh Cores

Salt marsh soil samples were collected by soil horizon for laboratory analysis of total carbon and total nitrogen content via method 1B1a1 [49] and analyzed via the dry combustion method 4H2a1-3a1 [49]. Laboratory analyses were processed at the USDA-NRCS Kellogg Soil Survey Laboratory in Lincoln, NE, as well as the University of Maryland Soils Laboratory.

### *2.4. Data Analysis*

To reflect spatial variability in $\delta^{13}C$ and $\delta^{15}N$ isoscapes, we grouped sample locations into four zones for data analysis. Grouping was based on sub-basins of the coastal lagoon system and considering hydrologically connected units, based on bathymetry, water residence time, and flushing characteristics described in Defne and Ganju [36]. Zone 1 (Barnegat Bay North) encompasses sites in northern Barnegat Bay until south of Toms River, where the depth is below 3 m, flushing occurs partially through Pt. Pleasant Canal, and water residence times largely exceed 15 days. Zone 2 (Barnegat Bay Inlet) stretches south to Manahawkin Bay, with depths up to 6 m, residence times typically less than one week, and flushing through Barnegat Inlet. Zone 3 (Little Egg Harbor) encompasses Little Egg Harbor, which is mostly shallow (<2 m) and poorly flushed by Barnegat Inlet (residence times can exceed 3 weeks). Zone 4 (Great Bay) encompasses all sampling locations in Great Bay and one station in the southeast of Little Egg Harbor, which is adjacent to a channel flushing through Little Egg Inlet. Zone 4 is mostly shallow (2 m) but has deeper channels, short residence times, and flushes through Little Egg Inlet. This regional grouping also corresponds to spatial variability in total N concentrations between June and November 1989–2009, with increased N concentrations in Zones 1 and 3 compared to Zones 2 and 4 [19]. We tested for significant differences in mean $\delta^{13}C$ and $\delta^{15}N$ between Zones using type III-ANOVA and Tukey's honest significant difference test where assumptions of normality and homoscedasticity were met. For non-normal sample populations with equal variances, we used Kruskal–Wallis test and Dunn's post hoc test.

Spatial interpolations of isotope data were conducted for combinations of species and isotopes that showed significant spatial autocorrelation, and for soil C density in the top 1 m of marsh sediments, soil C:N molar stoichiometric ratios, and water quality parameters, including both our data and publicly available data from the NJDEP [50]. Additionally, change of the C:N ratio in sediment cores over time was examined and spatial patterns

of the slope of the C:N ratio over time was mapped. Spatial interpolation was conducted using the ordinary kriging method, a spherical model semivariogram, and lag parameter based on the output raster cell size using ArcGIS 10.4 (ESRI, Redlands, CA, USA). The processing extent was constrained to the water and salt marsh areas around Barnegat Bay based on the estuarine emergent wetland classification from 2010 NOAA C-CAP land cover data [51].

## 3. Results

### 3.1. Modern Stable Isotopes

Stable isotope values measured in sediment, SPM, vegetation, and fauna where available from 35 sites in Barnegat Bay, New Jersey (Figure 3; Table S2) had an overall average $\delta^{15}N$ for all samples of $6.1 \pm 2.8‰$, while the average $\delta^{13}C$ value of all samples was $-16.8 \pm 4.3‰$ (n = 1147). Values of $\delta^{15}N$ for each sample type (Figure 3) had standard deviations ranging from 0.9‰ (*Z. marina*, n = 4 locations) to 3.7‰ (*P. australis*, n = 19 locations). Values of $\delta^{13}C$ for each sample type (Figure 3) had standard deviations ranging from 0.7 (*S. patens*, n = 27 locations) to 3.0‰ (*R. maritima*, n = 12 locations).

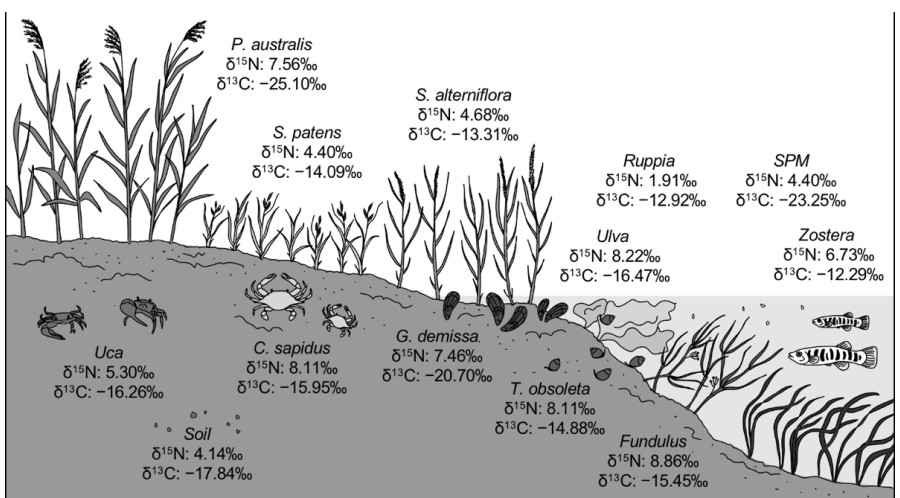

**Figure 3.** Species of primary producers, consumers, and particulates measured in this study. Values of nitrogen ($\delta^{15}N$) and carbon ($\delta^{13}C$) isotopes are averages between all measurements at 35 stations throughout Barnegat Bay. SPM stands for suspended particulate matter.

Significant differences in mean $\delta^{15}N$ across zones were found for SPM, primary producers (*P. australis*, *S. alterniflora*; Figures 4 and 5) and consumers (*C. sapidus*, *Fundulus* spp., *G. demissa*, *I. obsoleta*; Figure 5, Table S5). Significant differences in mean $\delta^{13}C$ between Zones were found for SPM, soil, and consumers (*Fundulus* spp., *G. demissa*; Figures 4 and 5; Table S5). While these differences were small (<1‰), post hoc tests revealed differences between mean $\delta^{15}N$ of Zones 1 and 2 (*Fundulus* spp., *S. alterniflora*), Zones 1 and 4 (SPM, *P. australis*), Zones 2 and 3 (SPM, *C. sapidus*), and Zones 3 and 4 (SPM, *G. demissa*). For $\delta^{13}C$, significantly different means (where differences were ≤1‰) were found for Zones 1 and 3 (SPM, *G. demissa*), Zones 1 and 4 (SPM, *Fundulus* spp., soil), and Zones 2 and 4 (SPM) (Figures 4 and 5; Table S5).

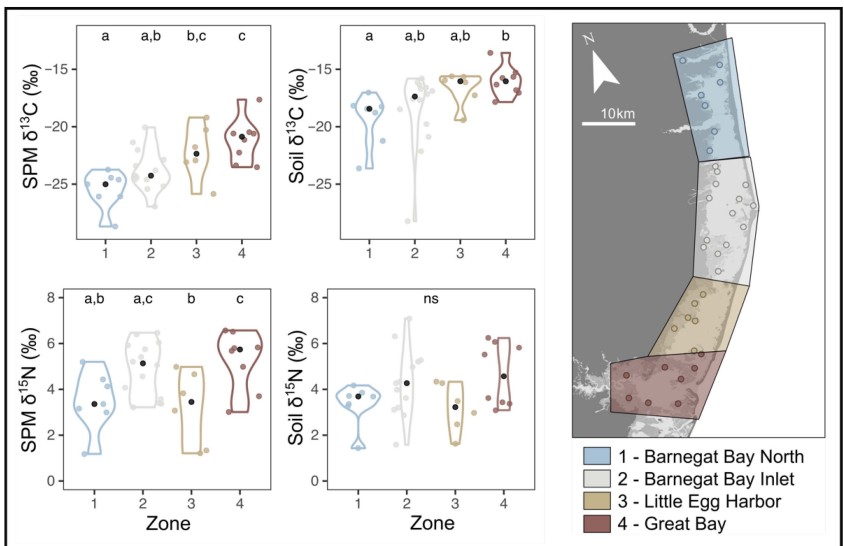

**Figure 4.** Distribution of suspended particulate matter (SPM) and salt marsh surface soil $\delta^{13}$C and $\delta^{15}$N across the four geographic zones that comprise the BB-LEH-GB. The map indicates sampling locations and zone boundaries. Letters within panels indicate differences between groups. For *p*-values and effect sizes, see Table S5.

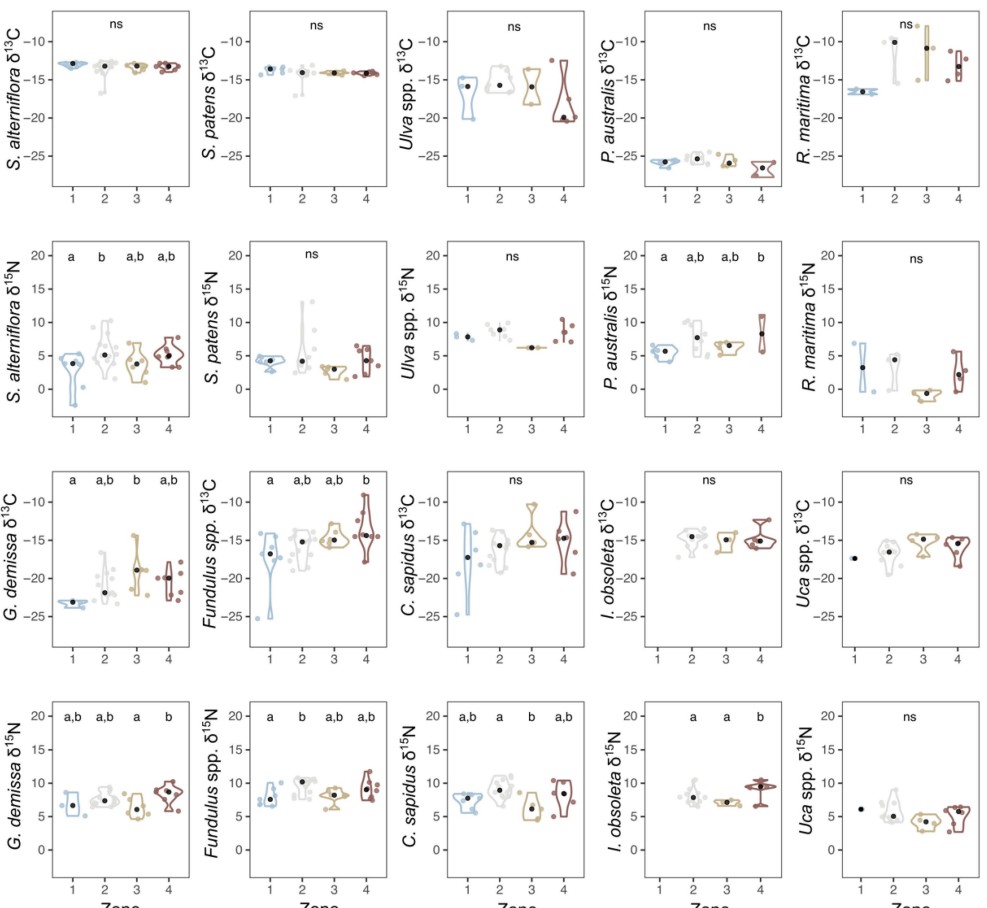

**Figure 5.** Distribution of $\delta^{13}$C and $\delta^{15}$N values (in ‰) for biota across the four geographic zones that comprise the BB-LEH-GB. Letters within panels indicate differences among groups, no significant difference is indicated by 'ns'. For *p*-values and effect sizes see supplementary Table S5.

Empirical Bayesian kriging was employed where the data had close to normal distributions based on the fit of histograms and Q–Q plots. Statistically significant spatial autocorrelation was observed in certain species based on a Global Moran's I test ($\delta^{15}$N *C. sapidus* z = 2.48, p = 0.013; $\delta^{13}$C *G. demissa* z = 2.23, p = 0.025). Spatial interpolations were mapped for biota (Figure 6), water quality parameters (Figures 7 and S1), CN stable isotope ratios of SPM (Figure S2), the downcore slope of C:N ratios of salt marsh soil cores (Figure S3), and the total C stock and C:N (molar) of the top 1 m of salt marsh soil cores (Figure S4).

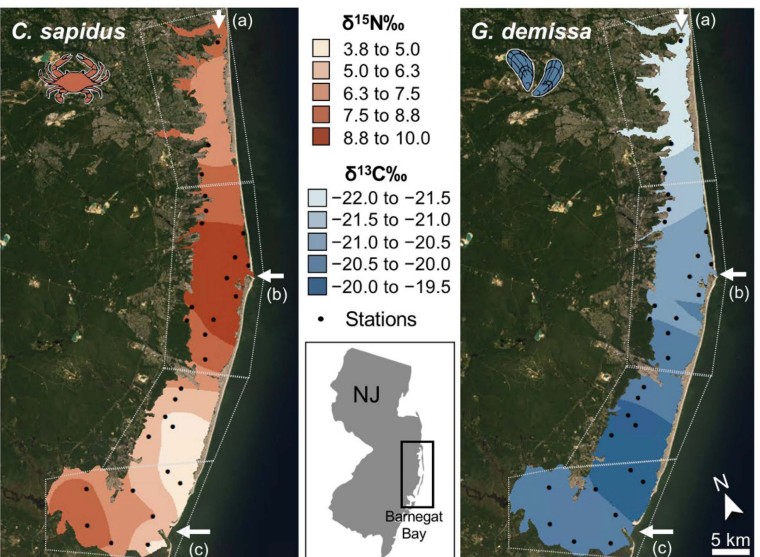

**Figure 6.** Interpolation maps based on Kriging in ArcGIS including $\delta^{15}$N in *C. sapidus* (**left**); and $\delta^{13}$C in *G. demissa* (**right**). Black points show locations of the 35 stations sampled. The interpolation extent is constrained to water and salt marsh areas around Barnegat Bay. White arrows show current marine inlets (**a**) Point Pleasant Canal; (**b**) Barnegat Inlet; (**c**) Little Egg Inlet. Imagery source: Esri World Imagery acquired in June 2020.

### 3.2. Archival and Modern Analogue Material

There were three species of archival plants available for analysis: *P. australis*, *S. alterniflora*, and *S. patens*, with collections ranging from the late 1800s to the early 1940s, before the widespread use of synthetic fertilizers (with their characteristic $\delta^{15}$N values of ~0 ‰). Historical plant specimens had an average $\delta^{15}$N value of 4.7 ± 1.6‰ (n = 8) and, when considered by species and zone, were isotopically enriched compared to modern samples, except in Zone 2 (Figure 8, Table S4). However, the limited number of biological replicates for museum specimens precluded statistical analysis.

One ribbed mussel (*G. demissa*) shell from 1962 was available from zone 2 and had a $\delta^{15}$N value of 14.8‰. Modern *G. demissa* shells had an average $\delta^{15}$N value of 7.6 ± 0.96‰ across all 25 sites. While the shell bound nitrogen values were significantly different from tissue values (p < 0.01, tissue $\delta^{15}$N averaged 6.3 ± 1.1 ‰), the values were not correlated (m = 0.13, r$^2$ = 0.1, data not shown).

*Ilyanassa obsoleta* shells collected between 1880 and 1975 had an average $\delta^{15}$N value of 12.6 ± 4.2‰ across all four zones. Following a slight $\delta^{15}$N enrichment after the late 1800s, values became significantly depleted through time when pooled across zones (p = 0.02). Modern collections of *I. obsoleta* shells had a mean $\delta^{15}$N value of 12.6 ± 2.4‰, which do not differ significantly from tissue values. Modern tissue and shell values were weakly correlated (m = 0.5473, r$^2$ = 0.2357, data not shown). We therefore consider shell and tissue isotopic data to be comparable for *I. obsoleta*.

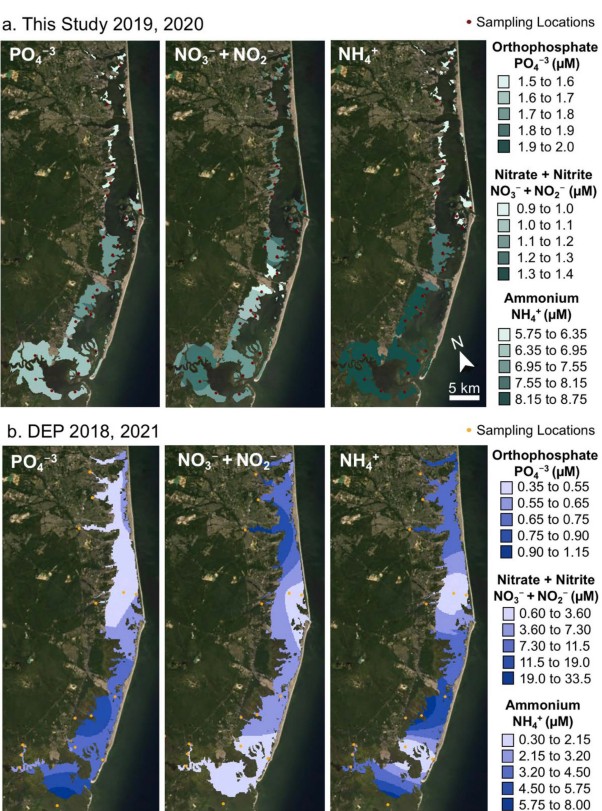

**Figure 7.** Interpolated (**a**) water quality data collected in this study from tidal marsh channels, and (**b**) open bay water quality samples collected in 2018 and 2021. For tidal marsh channels, ammonium values have a pronounced north to south gradient, with lower values in the north, and higher values in the south. For open water samples, the lowest values are found for dissolved inorganic nitrogen near the tidal inlets. For open water samples, phosphate shows a north to south gradient, with greater values in the south. Imagery source: Esri World Imagery acquired in June 2020.

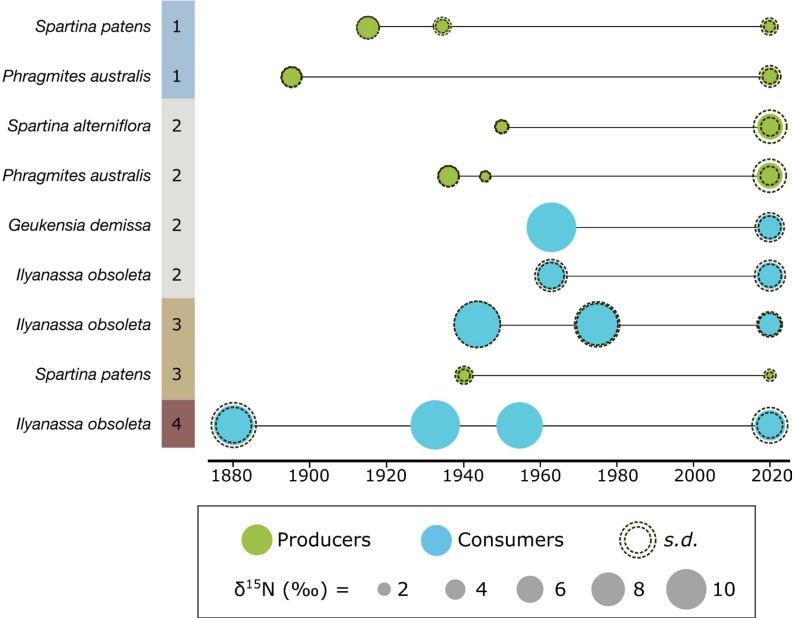

**Figure 8.** Comparison of $\delta^{15}N$ in historic and modern samples of producers (green) and consumers (blue), by zone (1 to 4). Shell-bound $\delta^{15}N$ was measured for historic consumer samples. Dashed circles indicate mean $\delta^{15}N \pm$ standard deviation, derived from technical replicates (historic specimens) or biological replicates (modern samples).

### 3.3. Water Quality and Salt Marsh Cores

Water samples collected from salt marsh channels for this study had orthophosphate concentrations ranging from 1.5 to 1.93 μM, with highest values at Zone 3 (Figure 7). Nitrate + nitrite ranged from 0.97 to 1.32 μM, with highest concentrations near the inlets (Zones 2 and 4) and low concentrations in Zone 3. Ammonium concentrations ranged from 5.75 to 8.78 μM and increased from north to south (Figure 7). Water quality monitoring samples collected from the bay water column at 14 stations in the summers of 2018 and 2021 [52] had orthophosphate concentrations ranging 0.34 to 1.13 μM, with highest concentrations in Zones 3 and 4 (Figure 7). Nitrate + nitrite concentrations ranged 0.64 to 33.5 μM, with highest concentrations in Zone 1 and lowest concentrations near the oceanic inlets. Similarly, ammonium concentrations ranged 0.38 to 7.99 μM with lowest values found near the oceanic inlets (Figure 7).

The profiles of salt marsh soil cores were annotated by horizon and analyzed for total C, total N, and % OC. Analysis of carbon stocks revealed spatial trends, with values double in the north (50–60 kg C m$^{-3}$) than in the south (20–30 kg C m$^{-3}$) (Figure S4; Table S6). In addition, sediment C:N ratios were lowest at Barnegat Inlet, with greater values elsewhere in the system (Figure S4). Sediment C:N ratios varied with depth in some sediment cores, with decreasing values found to the north of the BBLEH-GB estuary, and increasing values found in the southern part of estuary (Figure S3).

## 4. Discussion

Previous research has reported episodic algal blooms [53], low dissolved oxygen concentrations, and localized nutrient enrichment in BB-LEH-GB [39], raising concerns about eutrophication of the lagoon system [37]. However, the spatial heterogeneity of nutrient concentrations, patterns of primary productivity, and water residence times contributed to uncertainties regarding the eutrophic status of the embayment. Therefore, we assessed the C and N isoscapes at BB-LEH-GB to develop hypotheses about spatial and temporal patterns in nutrient availability, putative sources, and processing pathways and to provide a baseline for future studies of the BB-LEH-GB coastal lagoon system. To this end, primary producers, consumers, and suspended as well as sedimentary organic matter were analyzed for elemental composition and stable isotopic signatures; water quality parameters were tested at 32 sites. In addition, we compared C and N isotopic ratios of historic samples and their modern analogues. We found geographic and temporal patterns in the C and N isoscapes that we attribute to localized differences in macronutrient sources, concentrations, and processing driven by landuse and flushing characteristics of the estuary.

### 4.1. Spatial Patterns of Macronutrient Availability

BB-LEH-GB receives nutrient loadings primarily from nonpoint sources, such as surface water runoff, ground water, and atmospheric deposition. It is thought that stormwater runoff is the dominant mechanism for contribution of organic N while groundwater contributes historically accumulated dissolved nitrate [9]. Due to a larger area of urban landuse and larger inflow from the northern watersheds of Barnegat Bay, the concentrations and loadings of nitrate and total nitrogen in the water column are highest in the northern Zone 1 (Figure 7; [38,39,52]), with a second area of elevated nitrogen loading in Zone 3 at Mill Creek in Manahawkin Bay [19,54]. While loadings of phosphorus (P) follow this north–south gradient, the concentrations of P in the water column were found to be greatest in Little Egg Harbor (Zone 3; Figure 7; [39]). These opposing gradients in N and P concentrations result in differences in relative nutrient availability, with more P-limited conditions in the north (N:P = 88:1) and more N-limited conditions in the rest of the system (N:P = 18:1) [52].

In the north of Barnegat Bay, higher primary productivity, indicated by high chlorophyll-*a* concentrations (Figure S1), is supported by the availability of DIN and depleted P concentrations in the water column (Figure 7; [39,52]). In contrast, the reduced availability of nitrogen in the south limits primary productivity and P concentrations remain higher,

likely supplemented from oceanic water entering the bay and nutrient regeneration from the sediments and water column [52]. Although primary productivity was generally found to be lower in Zone 3 than northern sites (Figure S1; [52]), this region was repeatedly impacted by brown tide blooms [38,53]. It was suggested that drought conditions and the accompanying increases in salinity were conducive to algal blooms [53], likely exacerbated by longer residence times, which can reach 30+ days in the very north of Zone 1 and in Zone 3 [36]. Conversely, the shortest residence times can be found at the Barnegat and Little Egg Inlets, where water-column concentrations of both N and P are relatively low (Figure 7; [36,54]). We expected that N and C stable isotope ratios would reflect these complex spatial patterns in nutrient delivery, rates of primary productivity, and flushing, providing additional insight into the status of the system with regard to nutrient-enrichment and eutrophication [38,39].

### 4.2. Estuarine Isoscapes in the Region

Previous work from U.S. Northeast and mid-Atlantic estuaries have observed that $\delta^{13}C$ of sediment and particulate organic matter (POM) typically vary along a gradient from more depleted values upstream ($-25$ to $-27$ ‰) to less depleted near the marine inlets ($-20$ ‰; e.g., Plum Island Sound, [55]; Chesapeake Bay, [56]; Great Bay, NJ, USA, [57]). This pattern reflects differences in organic matter sources, with larger contributions of isotopically depleted upland (C3) plants upstream and less depleted marine organic matter downstream (ca. $-20$‰ in POM of the coastal Northwest Atlantic; [58]). In addition, it was shown that higher rates of primary productivity can increase $\delta^{13}C$ of organic matter [22,57].

Similarly, $\delta^{15}N$ of organic matter at upstream locations receiving terrestrial inputs are often less enriched ($-2$ to $2$‰) compared to downstream locations receiving marine organic matter (ca. $8$‰ in POM of the coastal Northwest Atlantic; [56–59]). However, this gradient in $\delta^{15}N$ is frequently altered by spatial variability in nutrient availability and processing. For example, $\delta^{15}N$ is enriched by high nitrogen loading, long residence times, and increased microbial processing (denitrification) in Long Island Sound [23], Chesapeake Bay [56], and Narragansett Bay [60]. These spatial isotope signals are generally not confined to primary producers and detritus, but also mirrored in consumers, which often display corresponding isoscapes [22,23,25,55,59].

### 4.3. Spatial Pattern in $\delta^{13}C$

In the BB-LEH-GB lagoon system, we found that the $\delta^{13}C$ isoscape shows a terrestrial-marine gradient similar to those observed in other estuaries of the region. We report a pronounced latitudinal gradient in the $\delta^{13}C$ of SPM, salt marsh soils, and some consumer species (Figure 4, Figure 5 and Figure S2), with the most depleted values in Zone 1 and the most enriched values in samples from Zone 4 (Figures 4 and 5). For SPM, we found samples from Zone 1 ($-25.5 \pm 1.6$‰) depleted relative to those from Zone 3 ($-22.2 \pm 2.3$‰) and Zone 4 ($-21.2 \pm 1.9$‰), with samples from Zone 2 ($-23.8 \pm 1.8$‰) depleted relative to those from Zone 4 (Figures 4 and 5). As SPM can comprise both plankton and detritus, the stable isotopes of SPM will reflect the net value of organic matter sources and in situ processing.

If SPM is composed primarily of plankton, SPM $\delta^{13}C$ can be indicative of productivity [22]: where higher $\delta^{13}C$ indicates more productivity in the water column. In addition, lower rates of productivity, which may result from less light availability in the subtidal, also result in the preferential uptake of $CO_2$ and associated lower $\delta^{13}C$ in many marine macrophytes [61]. In our study system, there is a latitudinal productivity gradient, with highest phytoplankton growth and chlorophyll-*a* concentrations in the north (Zone 1; Figure 2; [52]). However, this is also where $\delta^{13}C$ in SPM, soil, and some consumers (Figures 4–6) were most depleted, suggesting that rates of primary production do not drive the $\delta^{13}C$ isoscape of Barnegat Bay.

If SPM is composed mostly of detritus, SPM $\delta^{13}C$ can point to the relative contributions of different primary producers to the detrital pool. Organic matter producers performing

C3 photosynthesis have reduced $\delta^{13}C$ compared to those using C4 photosynthesis, illustrated by the much lighter $\delta^{13}C$ we report for *P. australis* (C3) organic matter compared to *S. alterniflora* and *S. patens* (both C4) tissues (Figures 3 and 5). This results in typical signatures of upland C3 plants of $-29‰$, while emergent wetland C4 vegetation has $\delta^{13}C$ of $-13‰$. As described above, plankton $\delta^{13}C$ can vary with $CO_2$ availability, but values reported in the literature are typically around $-21‰$ [62]. We suggest that the observed $\delta^{13}C$ gradient results from SPM in Zone 1 being dominated by terrestrial sources or *P. australis*, while the zones toward the south reflect an increasing influence of marine plankton and detritus from C4 salt marsh vegetation. This explanation is consistent with a larger, more developed watershed in the north and the much larger area of tidal wetlands in Zones 3 and 4 [63].

*4.4. Spatial Pattern in $\delta^{15}N$*

The $\delta^{15}N$ of organic matter also differed among regions, but instead of strictly following a terrestrial-marine gradient, the $\delta^{15}N$ isoscape follows the pattern determined by residence time. Samples from regions near oceanic inlets (Zones 2 and 4) show elevated $\delta^{15}N$ values compared to samples from Zones 1 and 3 (Figures 4–6). For example, SPM at Zone 3 ($3.2 \pm 1.6‰$) had depleted $\delta^{15}N$ relative to SPM from Zones 2 ($4.9 \pm 1.2‰$) and 4 ($5.3 \pm 1.3‰$), while SPM from Zone 1 ($3.5 \pm 1.3‰$) is depleted relative to SPM from Zone 4 (Figures 4 and S2). These patterns were likely the result of an interplay of variations in N sources and processing pathways. The observed SPM $\delta^{15}N$ was consistent with differences in source N, as samples from regions near inlets that provide oceanic N are enriched in $\delta^{15}N$ relative to sites that are more likely to receive their N from terrestrial and agricultural runoff (Zones 1 and 3).

Local differences in nutrient processing may also contribute to variation in the $\delta^{15}N$ isoscape. The highest N loads enter the study system in Zone 1 and at Mill Creek in Zone 3, which correspond to the sites of greatest $\delta^{15}N$ depletion in the samples we collected. Conversely, we report the highest $\delta^{15}N$ values in samples from presumably more N limited locations near Barnegat Inlet and Little Egg Inlet. Such a pattern is consistent with concentration-dependent fractionation, in which an abundance of N allows primary producers to selectively assimilate $^{14}N$, reducing $\delta^{15}N$ in organic matter. At N-limited locations, N-scarcity leads to an increased uptake of the less-favored $^{15}N$, resulting in $\delta^{15}N$ of organic matter close to source values. We suggest that concentration-dependent fractionation may contribute to the N isoscape of BB-LEH-GB, adding to recent evidence that this process alters the isoscape of estuarine food webs [64].

A further metabolic process altering $\delta^{15}N$ of organic matter is nitrogen fixation, which makes atmospheric N bioavailable, thereby introducing N with $\delta^{15}N$ of 0 ‰ into the system. This process is energetically costly and therefore conventionally assumed to occur mostly in N-limited systems, particularly where P is abundantly available [65,66]. At BB-LEH-GB, these conditions occur in Zone 3, where high P and low N availability combine with isotopically depleted $\delta^{15}N$, suggesting that nitrogen fixation could be a significant component of local N cycling. In addition, the process of denitrification is known to affect estuarine nitrogen pools, by selectively removing $^{14}N$ from the system, thereby enriching the $\delta^{15}N$ pool remaining for assimilation by primary producers. However, this process occurs predominantly where nitrogen is readily available; this is not the pattern we found of depleted $\delta^{15}N$ in regions with high N concentrations, like Zone 1. Interestingly, although this part of the estuary is the most urbanized, low $\delta^{15}N$ indicate that wastewater inputs are not a major source of N. In this regard, our findings contrast those from Long Island [23,67] or Narragansett Bay [41] where higher population density was found to covary with elevated $\delta^{15}N$ in estuarine biota. This finding attests to the efficiency of offshore discharge of effluents from sewage-plants at Barnegat Bay [38] and suggests little contribution of anthropogenic nonpoint source N to Zone 1.

We found similar nitrogen isoscapes of SPM (Figure S2) and emergent wetland plant tissue (*S. alterniflora*, *P. australis*; Figure 5), suggesting that primary producers draw from

the same N pool as plankton, generating similar isoscapes. Other primary producers examined in this study showed similar trends, but differences were not statistically significant, likely due to small sample sizes in our study. Our consumer N isoscapes are better supported and follow a similar pattern of elevated $\delta^{15}$N at Zones 2 and 4 relative to $\delta^{15}$N at Zones 1 and 3 (Figures 5 and 6). The ribbed mussel *G. demissa* feeds on plankton and detritus, while *Fundulus* spp. and *C. sapidus* are carnivores feeding on planktonic organisms, fishes and molluscs. Their stable isotopic composition reflects their food sources, mirroring spatial patterns of SPM $\delta^{15}$N, but with elevated values for all zones due to trophic enrichment. There are distinct isotopic baselines in different parts of BB-LEH-GB which are consistent across multiple trophic levels.

Overall, our analysis of nitrogen isoscapes suggests that $\delta^{15}$N of SPM, producers, and consumers differ spatially, with terrestrial runoff providing isotopically depleted $\delta^{15}$N to Zones 1 and 3, while oceanic nitrogen sources dominate at Zones 2 and 4. In addition, spatial variability in N processing potentially contributes to the observed nitrogen isoscape, with concentration-dependent fractionation contributing to more depleted $\delta^{15}$N in Zone 1 and nitrogen fixation potentially contributing to low $\delta^{15}$N at Zone 3.

*4.5. Temporal Trends in $\delta^{15}$N*

In addition to spatial variability in the C and N isoscapes of BB-LEH-GB, we analyzed historical samples from museum collections to investigate temporal trends in $\delta^{15}$N. Compared to modern analogues from the same locales, most historic samples had isotopically enriched $\delta^{15}$N (Figure 8). Assuming similar factors shape the modern and historic N isoscapes, we can attempt to interpret these temporal trends in $\delta^{15}$N. We suggest that at BB-LEH-GB, enriched $\delta^{15}$N indicates a larger influence of oceanic N sources, in contrast to lighter $\delta^{15}$N from terrestrial inputs. The decrease in $\delta^{15}$N of modern samples could indicate a reduction of oceanic exchange and decreased flushing in Zones 1 and 3 compared to samples from 1920–1980. However, flushing characteristics of BB-LEH-GB changed over this period (Figure 1), with the progressive infill of Barnegat Inlet from the 1940s to 1987 reducing the tidal prism by up to 60% [42]. Concurrently, the population at BB-LEH-GB grew exponentially and wastewater inputs into the Bay may have altered the N isoscape before the introduction of offshore discharge in the 1980s [38]. It is therefore conceivable that wastewater inputs of isotopically heavy nitrogen led to an enriched N isoscape, despite limited flushing and a reduction in heavier $\delta^{15}$N inputs from oceanic sources between 1940 and 1980. In addition, the more depleted $\delta^{15}$N in modern samples could reflect an increased contribution of fertilizer-derived N ($\delta^{15}$N ~0‰), although regulations on fertilizer application and an overall small fraction of agricultural use in the watershed suggest little fertilizer contribution to N loading.

For Little Egg Harbor (Zone 3), a decrease in $\delta^{15}$N over time has previously been reported in organic matter from sediment cores [68], supporting the trend we report from our analysis of museum specimens. We suggest that the modern, relatively depleted N isoscape is a result of nitrogen fixation in this N-limited part of BB-LEH-GB. This process could have become more important to the local N cycle, as sediment cores obtained for this study indicate a reduction in N deposition over time at Little Egg Harbor (Figure S3). Increased inputs of fixed N resulting in reduced $\delta^{15}$N would therefore be consistent with progressively higher N-limitation, which can be associated with increased diazotroph activity in salt marshes [69]. Moreover, an increased importance of terrestrial N sources due to reduced oceanic exchange and flushing of this part of the bay could have contributed to the decreasing trend of $\delta^{15}$N in soils and biota.

*4.6. Current Status of BB-LEH-GB in Relation to Nutrient Pollution*

The BB-LEH-GB estuary has been described as highly eutrophic [38] and water quality thresholds were violated in some portions of the bay for dissolved oxygen (Zone 3); turbidity (Zones 1 and 3); as well as N concentrations (Zone 1) when compared to thresholds set for other bays in the region [39]. Our study reinforces the notion that concerns over

water quality are most acute in the north of Barnegat Bay and at Little Egg Harbor; with the zones around oceanic inlets showing no signs of water quality impairment. Although our analysis of museum specimens points toward a historically more enriched N isoscape in the urbanized northern Barnegat Bay (Zone 1); wastewater discharge to the bay ceased four decades ago and a reduction in modern $\delta^{15}$N indicates that wastewater inputs and nutrient enrichment are largely decoupled in northern Barnegat Bay. Therefore; $\delta^{15}$N is not a strong indicator of water quality at BB-LEH-GB and should not be used for such purpose without careful consideration of its temporal and spatial variability in the system

Overall, nutrient enrichment, algal blooms, and low dissolved oxygen are primarily reported for those parts of BB-LEH-GB that have the longest water residence times and can be poorly flushed [36,39]. The urbanized north of Barnegat Bay (Zone 1) experiences the largest inputs of nutrients resulting in high phytoplankton growth [52], but only a small volume of water north of Toms River flushes through Point Pleasant Canal, while a larger volume of water exchanges through the distant Barnegat Inlet, resulting in long residence times of these nutrient-enriched waters [36]. Historically, this part of BB-LEH-GB harbored additional inlets (e.g., Cranberry Inlet 1758–1812; Beachhaven Inlet, 1850–1860s), potentially enhancing local flushing. Compared to modern samples, our museum specimen from zone 1 showed enriched $\delta^{15}$N in the early 20th century, when population density was low and we assume little wastewater contributed to N loadings. This could be indicative of efficient nutrient recycling in a still relatively oligotrophic system, where the relative contribution of oceanic N was larger than today. It appears likely that a BB-LEH-GB system with lower N loadings and better flushing would maintain more favorable water quality, particularly in the problematic northern portion at Tom's River. We suggest that any future alteration of hydrologic conditions of BB-LEH-GB, for example through inlet modification or tidal control structures [40], must carefully consider the importance of flushing and residence times of nutrient-enriched waters in the northern part of Barnegat Bay.

Lastly, we note that our ability to identify nutrient sources and processing pathways could have been improved by analyzing stable isotope ratios of dissolved nutrients in the water and atmospheric deposition of nitrogen through precipitation. While these measurements were outside the scope of our study, atmospheric deposition is potentially important to the N budget of Barnegat Bay [9] and knowledge of its isotopic composition would benefit future stable isotope assessments.

## 5. Conclusions

This study highlights the spatial heterogeneity of our study system with regard to macronutrient sources and processing pathways that are reflected in isoscapes of C and N. Stable N isotopes have routinely been used to assess nutrient pollution, with regard to wastewater as a source of excess nitrogen to coastal systems. We show how $\delta^{15}$N in coastal biota is context dependent and cannot be used as a wastewater indicator in isolation. Overall, the interpretation of C and N isoscapes of our study system required multiple lines of evidence, including macronutrient concentrations, hydrodynamic modeling, and sediment core data, emphasizing the need for careful consideration of spatial and temporal context when using stable isotope approaches for water quality assessments.

**Supplementary Materials:** The following supporting information can be downloaded at: https://www.mdpi.com/article/10.3390/w15010015/s1, Figure S1: Interpolations of water quality parameters measured in Barnegat Bay by NJ DEP for 2004-2021; Figure S2: Suspended particulate matter N and C isotope ratios; Figure S3: Change of the C/N ratio over core depth; Figure S4: Measurements of total soil carbon and C/N ratio in surface of marsh sediment cores; Table S1: Citations for historic maps and imagery; Table S2: Samples collected from Barnegat Bay and analyzed for stable C and N isotopes; Table S3: Citations for images used in Figure 2; Table S4: Historic and paired modern samples analyzed for C and N stable isotopes; Table S5: Results of tests for difference of mean $\delta^{15}$N and $\delta^{13}$C between regions; Table S6: Salt marsh soil core profiles, annotated by horizon.

**Author Contributions:** Conceptualization, A.J.O., E.B.W. and M.M.-D.; methodology, A.J.O., E.B.W., J.R.K., M.E.G., L.K.C. and D.S.; formal analysis, E.B.W., J.R.K., M.E.G., L.K.C., V.A., E.P. and M.J.S.; investigation, J.R.K., M.E.G., E.B.W. and A.J.O.; resources, E.B.W., A.J.O. and M.M.-D.; writing—original draft preparation, A.J.O., E.B.W., J.R.K., M.E.G., L.K.C. and D.S.; writing—review and editing, E.B.W., A.J.O., J.R.K. and M.E.G.; visualization, E.B.W., J.R.K. and L.K.C.; project administration, A.J.O., E.B.W. and M.M.-D.; funding acquisition, E.B.W., A.J.O. and M.M.-D. All authors have read and agreed to the published version of the manuscript.

**Funding:** This work was supported by Environmental Protection Agency under federal award numbers #83949701 and #CD9627340000-0. The statements, findings, conclusions, and recommendations are those of the authors and do not necessarily reflect the views of the US EPA and the funder played no role in study design, data interpretation, nor in the decision to submit the article for publication.

**Institutional Review Board Statement:** The Animal Care and Use Protocol was approved by the Institutional Animal Care and Use Committee of Drexel University (Protocol No 20783, 14 May 2019).

**Informed Consent Statement:** Not applicable.

**Data Availability Statement:** All data supporting the reported results are provided as Supplementary Materials.

**Acknowledgments:** We acknowledge Kirk Raper, Shannon Vasquez, Rob Tunstead, and Bronwyn Sayre for field assistance, Rick Lathrop for sharing historic land use data for Barnegat Bay, and Barbara Spinweber for administrative support. This is contribution #1513 from the Coastlines and Oceans Division of the Institute of Environment at Florida International University.

**Conflicts of Interest:** The authors declare no conflict of interest.

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
