# Peer review of "Tidal Flushing Rather Than Non-Point Source Nitrogen Pollution Drives Nutrient Dynamics in A Putatively Eutrophic Estuary"

_water, doi:10.3390/w15010015_

Round 1

Reviewer 1 Report

This manuscript studied the tidal flushing influenced nitrogen pollution and related nutrient dynamics in a New Jersey estuary (USA) by using the stable isotope method, which provided the significant environmental information of this field. Overall, this research is interesting and meaningful. Authors need to undergo Minor Revision before the paper can be accepted. The main remarks that in my opinion need attention are following:

1. Title: To me, the title is not well contained the “Water part”, but it is the choice of authors. It would be better to make it more concis.

2. Abstract: please add an indication of the achievements in water research field from your study. Moreover, some key data should be incorporated into the abstract.

3. Introduction: The introduction should more clearly show the knowledge gaps identified and link them to the paper goals within 4~5 paragraphs (succinct and clear).

4. Discussion: If more isotope data of atmospheric deposition (e.g, δ15N of rainwater nitrate) are analyzed here, it will support your discussion better.

Reviewer 2 Report

In the paper, the authors describe their research related to the degree of anthropogenic pollution in Barnegat Bay (New Jersey, USA). A series of studies providing data on the concentration of phosphates and nitrates in water as well as the content of stable carbon and nitrogen isotopes in a number of organisms - both flora and fauna, are presented here. The article is well written, very interesting and should be published. A few of the following minor comments should be taken into account by the authors in the final version of the article:

1. Lack of information on 13C and 15N deltas in water samples - did the authors measure the isotopic composition for phosphates and nitrates dissolved in water?

2. The definition of isotopic delta should rather be found in the introduction (where delta appears for the first time).

3. What do the letters a, b, c, ns mean in figures 4 and 5?

Reviewer 3 Report

Dear author(s),

there are some inspiring insights thorough the manuscript and I tend to agree on its publication. However, there are few points that needs to be quickly addressed to improve its overall communication:

Title:

1/ better address our international audience of readers, do not indicate local impact

2/ clearly condensate the novelty and significance of the main discovery into a short and groundbreaking claim

Abstract:

3/ strictly follow the established schema of writing academic Abstract: A/ introduction (urgency and significance of the research hypothesis); B/ principles of the methods used + key results; C/ conclusions (commercial and environmental impacts)

4/ better highlight the global impact and urgency of your findings, clearly indicate how will our international audience benefit from these revelations (results limited to "Barnegat Bay" needs deeper synthesis and generalization if you want to get published on international level)

Introduction:

5/ remove all clusters of references to avoid reference overkill (prefer only 1 reference to support 1 claim)

6/ anthropogenic impacts should be discussed more broadly, refer to paper "Techno-economic review on short-term anthropogenic emissions of air pollutants and particulate matter"

7/ go straight to the point and more in depth, write more technically (always provide corresponding numbers), significantly condensate all the text by reducing local issues, ballast phrases and cliché

8/ deeper review the latest trends in nutrient management, refer to papers "Economic impacts of soil fertility degradation by traces of iron from drinking water treatment" and "Novel sorbent shows promising financial results on P recovery from sludge water"

9/ make sure that this chapter fully introduces any reader into to the topic, explain all the terms, units, abbreviations, Latin and Greek letters, and the whole context that is necessary for anyone (including experts from other disciplines) to understand the following chapters

10/ do not ignore (economic) reality, comment on the financial aspects linked with N and C sources, refer to papers "Does the life cycle affect earnings management and bankruptcy?" and "Economic considerations on nutrient utilization in wastewater management"

11/ the research hypothesis could be stated more clearly, condensate the research hypothesis into 1 short statement or question that will be subsequently confirmed or refuted, make sure the urgency and significance of the research hypothesis was justified in its environmental - economic nexus

Materials and Methods:
12/ do not present encyclopedic information here, present only the methodology (procedures) - in such a way that it can be reproduced anytime, by anyone, anywhere (do not create obstacles like referring to specific location etc.)

13/ please understand that the methodology must be described in a completely unambiguous way that does not allow for multiple interpretations (everyone who reads this chapter should get very precise instructions on how to repeat your procedure to achieve exactly the same results)

14/ each material, reactant, apparatus and procedure used needs to be presented in detail (serial number, setup, process parameters, manufacturer, country of origin, purity etc.)

15/ Fig. 1: kindly understand that international audience of readers can not understand notes such as "NJ fertilizer law passed" it is necessary to raise the communication of your manuscript to a scientific level = be more explanatory

16/ Fig. 2: it looks like you are presenting some "results" here, if so, them move it into the chapter "Results", kindly note that the chapter "Materials and Methods" should contain only detailed instructions how to replicate your methods

Results:

17/ do not use decimal symbol when communicating %, "6.1%" = 6% etc.

18/ Fig. 4: make sure each axis of each chart is provided with corresponding units

19/ avoid data overkill, present only the most most industrially and environmentally important results

20/ please understand that no (location) place names provide information value to readers, you can bet that almost none of them have ever heard of the case study location, it's always enough to just enter the GPS coordinates

21/ Fig. 6 and 7: make sure you have permission (from Google?) to publish these maps

Discussion:

22/ show more self-criticism to your work (can all the methods and results be fully trusted? what are the weaknesses of the methods used? where do the main measurement inaccuracies arise? what are the limitations ? are the lessons learned transferable to other fields?)

22/ I recommend to discuss in more depth the acceptability (digestibility  = bioavailability) of different forms of C and N to living creatures, refer to paper "Advances in nutrient management make it possible to accelerate biogas production and thus improve the economy of food waste processing"

23/ compare your results in more depth with the existing literature, identify the main deviations and try to explain the mechanisms by which they may have been caused

24/ the economic and social implications should not be ignored

25/ reveal the main driving mechanisms of your results, provide deeper synthesis and reveal some more original/significant findings (higher level of generalization is advisable)

Conclusions:

26/ do not repeat your methods and "results" again and again, please understand that the "Conclusion" chapter is not a summary of your work, present only original and industrially significant revelations that have the potential to expand the horizon of human knowledge (higher level of generalization is mandatory = not limited to the case study)

27/ clearly indicate whether the research hypotheses tends to be confirmed or not and highlight the significance of your work

Round 2

Reviewer 3 Report

My comments were ignored, I can not give my green light to publication of the manuscript in its current stage.